# Optimal testing using combined test statistics across independent studies

**Lasse Vuursteen**[*]
Delft Institute of Applied Mathematics
Delft University of Technology
`l.vuursteen@tudelft.nl`

**Botond Szabó**
Department of Decision Sciences
and Institute for Data Science and Analytics
Bocconi University
`botond.szabo@unibocconi.it`

**Aad van der Vaart**
Delft Institute of Applied Mathematics
Delft University of Technology
`a.w.vandervaart@tudelft.nl`

**Harry van Zanten**
Mathematics Department
Vrije Universiteit Amsterdam
`j.h.van.zanten@vu.nl`

## Abstract

Combining test statistics from independent trials or experiments is a popular method of meta-analysis. However, there is very limited theoretical understanding of the power of the combined test, especially in high-dimensional models considering composite hypotheses tests. We derive a mathematical framework to study standard meta-analysis testing approaches in the context of the many normal means model, which serves as the platform to investigate more complex models.

We introduce a natural and mild restriction on the meta-level combination functions of the local trials. This allows us to mathematically quantify the cost of compressing $m$ trials into real-valued test statistics and combining these. We then derive minimax lower and matching upper bounds for the separation rates of standard combination methods for e.g. p-values and e-values, quantifying the loss relative to using the full, pooled data. We observe an elbow effect, revealing that in certain cases combining the locally optimal tests in each trial results in a sub-optimal meta-analysis method and develop approaches to achieve the global optima. We also explore the possible gains of allowing limited coordination between the trial designs. Our results connect meta-analysis with bandwidth constraint distributed inference and build on recent information theoretic developments in the latter field.

## 1 Introduction

Given multiple data sets relating to the same hypothesis, one would like to combine the evidence. Sometimes, the full data sets are not available (e.g. due to privacy or proprietary reasons) or difficult to combine directly (e.g. due to the different experimental or observational setups). In such cases, the analysis must be carried out on the basis of the published results for each of the studies. Such "meta-analysis" can increase the statistical power by combining individually inconclusive or moderately significant tests, while keeping the false positive rate under control. Therefore, meta-analysis has received a lot of attention in various fields, for instance in genetics and system biology, when studying rare variants [4, 16] or in deep learning, for few shot image recognition and neural architecture search, see the review article [24].

---

[*]

The outcomes of the studies concerning hypothesis tests are, typically, summarized as real-valued test statistics and/or associated p-values. One expects the combination of $m$ such p-values to result in an increase in power, but one also expects to pay a price relative to computing a test on the basis of the full, pooled data of the $m$ trials. The question of how to optimally combine independent real-valued test statistics concerning the same hypothesis into a single test has an extensive literature. A multitude of methods for combining independent tests of significance exist. For combining p-values, this starts with Fisher, Tippett and Pearson in the nineteen-thirties, see [42, 18, 32, 36, 21, 28, 45, 14, 31, 50, 48, 12, 47] and references therein. In Section 3, we collect and describe the most popular and frequently used p-value combination techniques.

As noted in [9], there does not exist a general uniformly most powerful p-value combination method for all alternative hypotheses. The distribution of a p-value or its underlying test statistic under the alternative hypothesis should be taken into consideration when selecting a method of combination. The performance of different p-value combination techniques was investigated extensively by empirical experiments in various synthetic and real world scenarios, see for instance [29, 51]. However, a unified, general theoretical description is lacking, especially in non-trivial, multi-dimensional composite testing problems, where the likelihood ratio test is not necessarily uniformly most powerful.

E-values are an increasingly popular and important notion of evidence, see [35, 22, 34]. E-values allow the combination of several tests in a straightforward manner while preserving the prescribed level of the tests (see Section 3.2). Formally, e-values are nonnegative random variables whose expected values under the null hypothesis are bounded by one. In contrast to p-values defined by probabilities, e-values are defined by expectation. This imposes significant differences in their interpretation, application and combination compared to the more standard p-values. However, as for p-values, very little is known about the power of these combination procedures. Theoretical results focus on specific optimality criteria, for instance the worst-case growth-rate (GROW), see [22]. However, these do not directly imply guarantees on the testing power, which is the main focus in practice.

Our focus is on multidimensional models, where a certain loss in power is to be expected, since combining multidimensional data into a real-valued statistic (e.g. p-value or e-value) requires data compression. Typically, summary statistics are combined by some "reasonable" function $C_m : \mathbb{R}^m \to \mathbb{R}$, where "reasonable" means that $C_m$ should not exploit the richness of the real numbers to encode the data in full. We aim to quantify the loss of summarizing, the gain of performing a meta-analysis and the best testing strategies in the individual experiments meta-analysis.

We consider the signal detection problem in the many normal means model, see Section 2 for the detailed description. One possible interpretation of this testing problem is to learn whether a treatment has an effect on any of the dimensions investigated. This model is directly applied in several fields where high-dimensional statistics and machine learning settings are concerned, such as detecting differentially expressed genes [33, 27, 30, 41, 15], bankruptcy prediction for publicly traded companies using Altman's Z-score in finance [5, 6], separation of the background and source in astronomical images [13, 23], and wavelet analysis [1, 26]. Furthermore, the model allows for tractable computations and it typically serves as the platform to investigate more difficult statistical and learning problems, including high- and infinite-dimensional models, see for instance [25, 43, 15, 20]. In each experiment $j \in \{1, ..., m\}$ the observations are summarized by an appropriate real-valued summary statistic $S^{(j)}$. These local test statistics (e.g. p- or e-values) are combined into $C_m(S^{(1)}, \ldots, S^{(m)})$. We consider a general class of combination functions $C_m$, requiring only Hölder type continuity. This introduces only a mild restriction, and includes many standard meta-analysis techniques, for instance the standard p-value combination methods (see Section 3.1); e-value techniques (see Section 3.2); and other ad hoc and natural test statistic combination approaches, see the beginning of Section 3 for additional examples.

Our setting provides a principled and unified framework to study the power of standard meta-analysis testing methods. Within the framework of the many normal means model, we derive a minimax lower bound for the testing (separation) error and provide test statistics with associated combination methods that attain this theoretical limit (up to a logarithmic factor). Our results reveal that there is a certain unavoidable loss associated with compressing the data of each experiment to a real valued test statistic. We see that while it is always possible to obtain better testing rates using $m$ trials instead of the best possible test based on a single trial, there is always a loss incurred when compared to the

full, pooled data and optimal test in moderate- to large dimensional problems. Our theoretical results quantify these gains and losses in terms of the dimension $d$, sample size $n$ and number of trials $m$.

Furthermore, we observe an elbow effect, which occurs when the number of trials is large compared to the dimension of the signal. In this regime, combinations of the (locally) optimal test in each individual trial performs sub-optimally as a whole when aggregated and meta-analysis approaches based on directional test statistics are shown to perform better. Finally, we show that the performance of the meta-level tests can substantially improve (in certain regimes, depending on $d, m, n$) if a certain amount of coordination between the trials is allowed (e.g. by having access to the same random seed). For the theoretical analysis of meta-analysis techniques we derive connections with the distributed statistical learning literature under communication constraints. Our paper builds on the recent information theoretical developments in distributed testing [2, 39, 40], allowing us to address several fundamental questions for the first time with mathematical rigor.

The paper is organised as follows. In Section 2 we introduce the mathematical framework we consider in our investigation and present the corresponding minimax testing lower bound results. Next in Subsection 2.1 we show that the derived results are sharp by providing several meta-analysis approaches attaining the limits. Then we investigate the benefits of allowing a mild coordination between the trials in Subsection 2.2. We collect and discuss the standard p- and e-value combination methods in Section 3 and demonstrate our theoretical results numerically on synthetic data sets in Section 4. We discuss our results and derive conclusions in Section 5. The proofs of our results are deferred to the Appendix. In Section A.1 we present the proof of our main results while the proofs of the technical lemmas are given in A.2.

**Notation:** For two positive sequences $a_n, b_n$ we write $a_n \lesssim b_n$ if the inequality $a_n \leq C b_n$ holds for some universal positive constant $C$. Similarly, we write $a_n \asymp b_n$ if $a_n \lesssim b_n$ and $b_n \lesssim a_n$ hold simultaneously and let $a_n \ll b_n$ denote that $a_n/b_n = o(1)$. Furthermore, we use the notations $a \vee b$ and $a \wedge b$ for the maximum and minimum, respectively, between $a$ and $b$. Throughout the paper $c$ and $C$ denote global constants whose value may change from one line to another.

## 2 Main results

In our analysis, we consider the localized version of the many normal means model, tailored to investigating meta-analysis techniques. We assume that in each local trial or experiment $j \in \{1, ..., m\}$ we observe a $d$-dimensional random variable $X^{(j)} \in \mathbb{R}^d$, subject to

$$X^{(j)} = f + \frac{1}{\sqrt{n}} Z^{(j)}, \qquad Z^{(j)} \overset{iid}{\sim} N(0, I_d), \quad j = 1, ..., m, \tag{1}$$

for some unknown $f \in \mathbb{R}^d$. We denote by $\mathbb{P}_f$ the joint distribution of the observations and let $\mathbb{E}_f$ be the corresponding expectation. We note that this framework is equivalent to having $n$ independent $N(f, I_d)$ observations within each local sample.

Our goal is to test the presence or absence of the "signal component" $f \in \mathbb{R}^d$. More formally, we consider the simple null hypothesis $H_0 : f = 0$ versus composite alternative hypothesis $H_\rho : \|f\|_2 \geq \rho$, for some $\rho > 0$. This corresponds to testing for joint significance of variables, such as the presence of an effect of a treatment on any of the dimensions investigated. The difficulty in distinguishing the hypotheses depends on the effect size, the sample size and the dimension $d$. Here, $\rho$ can be seen as the smallest effect size deemed important.

For a $\{0, 1\}$-valued test $T$, define the testing risk $\mathcal{R}(T, H_\rho)$ as the sum of the Type I error probability and worst case Type II error probability, i.e.

$$\mathcal{R}(T, H_\rho) := \mathbb{P}_0(T = 1) + \sup_{f \in H_\rho} \mathbb{P}_f(T = 0). \tag{2}$$

In the case of a single trial (i.e. $m = 1$), this testing problem is known to have minimax separation rate or "detection boundary" $\rho^2 \asymp \sqrt{d}/n$.

This means that if $\rho^2 \gg \sqrt{d}/n$, there exist consistent[2] tests $T \equiv T_{d,n}$ in the sense that $\mathcal{R}(T, H_\rho) \to 0$, whilst no consistent tests exist when $\rho^2 \ll \sqrt{d}/n$. That is, for effect sizes of smaller order than

---

[2] For any asymptotics in $\rho$, $d$ and $n$ such that $\rho^2 \gg \sqrt{d}/n$.

$\sqrt{d}/n$, the null hypothesis cannot be consistently distinguished from the alternative hypothesis. Such a testing rate is attainable through a chi-square test based on $\|\sqrt{n}X^{(1)}\|_2^2$ (see e.g. [7]).

In case of $m$ trials, if the full data were pooled (with aggregated sample size $nm$), the minimax separation rate would be $\sqrt{d}/(mn)$. However, pooling the data might not be possible or allowed in practice and often only real-valued test statistics are available that describe the significance in the local problems (e.g. a p- or an e-value). These $m$ test statistics $S^{(j)}$, $j = 1, ..., m$, then can be combined with some combination function $C_m : \mathbb{R}^m \to \mathbb{R}$, providing the test statistic in the meta-analysis. We now ask whether the above pooled testing rate is attainable with this meta-analysis procedure.

Without any restrictions on the test statistics $S = (S^{(1)}, \ldots, S^{(m)})$ or the combination function $C_m$, any of the conventional optimal "full-data" tests can be reconstructed, since the real numbers and mappings between the real numbers form an overly rich class. We wish to restrict our analysis to $S$ and $C_m$ that are reasonable in practice and capture (most of) the relevant meta-analysis methods as listed in Section 3.

Based on each of the local observations $X^{(j)}$, a real-valued test statistic $S^{(j)}$ is computed, where each $S^{(j)}$ is a function of $X^{(j)}$ and possibly a source of randomness $U^{(j)}$ independent of $X :=$ $(X^{(1)}, \ldots, X^{(m)})$.

**Assumption 1.** *For measurable functions $f_j : \mathbb{R}^d \times \mathbb{R} \to \mathbb{R}$ and independent random variables $U^{(1)}, \ldots, U^{(m)}$ which are independent of the data $X$, the $j$-th test statistic $S^{(j)} = f_j(X^{(j)}, U^{(j)})$ satisfies $\mathbb{E}_0|S^{(j)}| \leq M$, for some $M > 0$, $j = 1, \ldots, m$.*

We consider Hölder continuous combination functions $C_m : \mathbb{R}^m \to \mathbb{R}$. Arguably, this is the most important assumption in ruling out bijections between $\mathbb{R}^d$ and $\mathbb{R}$. This ensures that a small change in the underlying local test statistics cannot result in a large change in the combination of test statistics $C_m(S^{(1)}, \ldots, S^{(m)})$.

**Assumption 2.** *There exist $L, p, q > 0$ such that for all $s, s' \in \mathbb{R}^m$*

$$|C_m(s) - C_m(s')| \leq L \Big( \sum_{j=1}^m |s_j - s'_j|^p \Big)^q. \tag{3}$$

The special case of $p = 2$ and $q = 1/2$ leads to Lipschitz continuous functions. Assumption 1 and Assumption 2 should be considered in conjunction. By rescaling and centering test statistics $S^{(j)}$, one can typically obtain test statistics satisfying Assumption 1. Rescaling and centering typically does affect how the test statistics need to be combined, which might "break" Assumption 2.

Finally, following the standard testing approach, we compare the aggregated test statistics $C_m(S^{(1)}, \ldots, S^{(m)})$ to a threshold value. If the combined test statistics result in a large enough value, the null hypothesis of no effect is rejected. We note here that two sided tests can be written as one-sided tests through straightforward transformations (e.g. centering and taking absolute value). More formally, we consider tests $T_\alpha$ of level $\alpha$ satisfying the following assumption.

**Assumption 3.** *There exists a strictly decreasing function $\alpha \mapsto \kappa_\alpha$ so that*

$$T_\alpha = \mathbb{1}\Big\{ C_m(S^{(1)}, \ldots, S^{(m)}) \geq \kappa_\alpha \Big\} \tag{4}$$

*satisfies $\mathbb{P}_0 T_\alpha \leq \alpha$.*

The map $\alpha \mapsto \kappa_\alpha$ could be taken as the quantile function of $C_m(S^{(1)}, \ldots, S^{(m)})$ under its null distribution if it is appropriately standardized. If $\mathbb{E}_0 C_m(S^{(1)}, \ldots, S^{(m)})$ is bounded in $m$, we can choose $\kappa_\alpha$ equal to $1/\alpha$ times the upper bound, in view of Markov's inequality.

Our first main result, Theorem 1 below, establishes a lower bound for tests of the form (4) and $C_m$ and $S$ satisfying the above assumptions. More concretely, under our assumptions, any test $T_\alpha$ (of level $\alpha \leq 0.1$) has large Type II-error under alternatives with $\rho^2$ of smaller order than $(\sqrt{m} \wedge \frac{d}{\log(m)})\sqrt{d}/(mn)$. When the number of trials is small compared to the dimension (i.e. $m \log^2(m) \leq d^2$), this means that the separation rate is at least $\sqrt{d}/(\sqrt{m}n)$. Thus even though there is a benefit in terms of separation rate compared to testing based on just a single trial, the

gain is at best the square root of what one would gain based on testing on the pooled data. When $m \log^2(m) \geq d^2$, the rate in the lower bound changes to $d\sqrt{d}/(mn \log(m))$, resulting in an elbow effect.

**Theorem 1.** *Let $S^{(1)}, \ldots, S^{(m)}$, $C_m$ and $T_\alpha$ satisfy Assumptions 1–3 with $T_\alpha$ of level $\alpha \in (0, 0.1]$. Then there exists a constant $c > 0$ depending only on $L$, $p$, $q$ and $M$, such that if*

$$\rho^2 \leq c \frac{(\sqrt{m} \wedge \frac{d}{\log(m)})\sqrt{d}}{mn}, \tag{5}$$

*it holds for all $n, m, d \in \mathbb{N}$ that*

$$\sup_{f \in H_\rho} \mathbb{P}_f (T_\alpha = 0) \geq 3/4. \tag{6}$$

*Remark* 1. The ranges of values $0 < \alpha \leq 0.1$ and $\beta = 3/4$ for the Type I and II errors, respectively, are arbitrary. Similar results hold for different choices as well. For instance, one can take arbitrary $\alpha \in (0, 1/5]$ and $\beta \in (0, 2/3]$, see the proof of the theorem for details. The result implies in particular that consistent testing is not possible for signals of a smaller order than the right hand side of (5), where asymptotics can be considered in $n$, $m$ and $d$ simultaneously.

In the next section we show that the lower bounds in the theorems above are sharp (up to a logarithmic factor).

## 2.1 Rate optimal combination methods

To attain the lower bound rate derived in Theorem 1, different tests can be considered. The optimal rate displays an elbow effect around $m \asymp d^2$. When the dimension is large compared to the number of trials $m$ (i.e. $m \lesssim d^2$), strategies that combine p-values for the optimal local tests (based on $\|\sqrt{n}X^{(j)}\|_2^2 \sim^{H_0} \chi_d^2$), turn out to achieve the optimal rate, as exhibited below. Such a test statistic is invariant to the directionality of $X^{(j)}$ and invariant under the model in the sense that the resulting power for the alternative $\mathbb{P}_f$ or $\mathbb{P}_g$ is the same as long as $\|f\|_2 = \|g\|_2$.

On the other hand, when the dimension is small compared to the number of trials (i.e. $m \gtrsim d^2$), optimal strategies exhibited below use information on the direction of $X^{(j)}$. In fact, we show in Theorem 4 in the Appendix that if no such information is available (i.e. the events defined by the signs of the $(X^{(j)})_{j=1,\ldots,m}$ vector are not contained in the sigma algebra generated by the test statistics $S$), one cannot obtain a rate better than $\sqrt{d}/(\sqrt{m}n)$. This implies that by combining the locally optimal test statistics $S^{(j)} = \|\sqrt{n}X^{(j)}\|_2^2$ (or their arbitrary functions, e.g. the corresponding local p-values) would result in information loss and hence sub-optimal rates in the meta-analysis.

Furthermore, it turns out, in accordance with the empirical literature discussed in the introduction, that there does not exist a uniquely best meta-analysis method. In fact, multiple standard meta-analysis techniques provide (up to a logarithmic factor) optimal rates, see below for some standard approaches attaining the lower bounds derived in Theorem 1.

First we consider the scenario when the dimension $d$ of the model is large compared to the number of trials $m$, i.e. $m \lesssim d^2$. Locally the optimal test is based on the test statistic $\|\sqrt{n}X^{(j)}\|_2^2 \overset{H_0}{\sim} \chi_d^2$. A natural way to combine these statistics would be to sum these locally optimal test statistics to obtain

$$T_\alpha = \mathbb{1} \left\{ \sum_{j=1}^m \left\| \sqrt{n}X^{(j)} \right\|_2^2 \geq F_{\chi_{dm}^2}^{-1} (1 - \alpha) \right\}, \tag{7}$$

which has level $\alpha$. Alternatively, one could also apply p-value combination methods, such as Fisher's or Edgington's method based on the p-value $p^{(j)} = 1 - F_{\chi_d^2}(\|\sqrt{n}X^{(j)}\|_2^2)$, see Section 3. Lemma 6 in the appendix establishes that these tests are rate optimal.

Second, consider the case that the number of trials is large compared to the dimension, i.e. $m \gtrsim d^2$. Rate optimal tests can be constructed based on a variation of Edgington's or Stouffer's method, see Section 3 for their descriptions. Taking a partition of $\{1, \ldots, m\} = \cup_{i=1}^d \mathcal{J}_i$ where $|\mathcal{J}_i| \asymp m/d$ and

setting $S^{(j)} = \sqrt{n} X_i^{(j)}$ if $j \in \mathcal{J}_i$, the meta-level test

$$T_\alpha = \mathbb{1}\left\{ \frac{\sqrt{d}}{m} \sum_{i=1}^d \left( \sum_{j \in \mathcal{J}_i} S^{(j)} \right)^2 \geq d^{-1/2} F_{\chi_d^2}^{-1}(1-\alpha) \right\} \tag{8}$$

achieves the lower bounds. The above test is similar to employing Stouffer's method for each of the coordinates and averaging, i.e. computing approximately $m/d$ iid p-values $p^{(j)} = \Phi(\sqrt{n} X_i^{(j)})$ for $j \in \mathcal{J}_i$ and applying the inverse Gaussian CDF $\Phi^{-1}(p^{(j)})$. Alternatively, the following variation of Edgington's method,

$$T_\alpha = \mathbb{1}\left\{ \frac{\sqrt{d}}{m} \sum_{i=1}^d \left( \sum_{j \in \mathcal{J}_i} \left( p^{(j)} - \frac{1}{2} \right) \right)^2 \geq \kappa_\alpha \right\}, \tag{9}$$

is also rate optimal, as proven in Lemma 7 in the appendix. Essentially, these strategies divide the trials accross the $d$ different directions, and combines the evidence for each of the directions. Theorem 4 affirms that the information on the "direction" of the data is crucial to achieve the optimal rate in the $m \gtrsim d^2$ case, by showing that strategies that do not contain such information (rotationally invariant strategies such as norm-based test statistics) achieve the rate $\sqrt{d}/(\sqrt{m}n)$ at best. We summarize the above testing upper bounds in the theorem below.

**Theorem 2.** *For all $\alpha, \beta \in (0,1)$ there exist $S$, $C_m : \mathbb{R}^m \to \mathbb{R}$ and tests $T_\alpha$ of level $\alpha$ satisfying Assumptions 1–3 such that if*

$$\rho^2 \geq C_{\alpha,\beta} \frac{(\sqrt{m} \wedge d)\sqrt{d}}{mn}, \tag{10}$$

*we have*

$$\sup_{f \in H_\rho} \mathbb{P}_f \left( T_\alpha = 0 \right) \leq \beta$$

*for a large enough constant $C_{\alpha,\beta} > 0$ depending only on $\alpha, \beta \in (0,1)$, for all $n, m, d \in \mathbb{N}$.*

## 2.2 Benefits of coordination between the trials

When the dimension is small relative to the number of trials, as exhibited in the previous section, optimal strategies include information on the directionality of the observation vector. In this section we show that in this regime, there could be an additional benefit from allowing mild coordination between the trials through employing shared randomness, e.g. a shared random seed between the trials. Such a phenomenon has been observed before in the distributed testing literature [3, 2, 39, 40], which forms the basis of our analysis below.

We consider the following variation on Assumption 1, where the key difference is that the source of randomness is allowed to be shared between the $m$ trials.

**Assumption 4.** *For functions $f_j : \mathbb{R}^d \times \mathbb{R} \to \mathbb{R}$ and a random variable $U$ which is independent of the data $X$, the $j$-th test statistic $S^{(j)} = f_j(X^{(j)}, U)$ satisfies $\mathbb{E}_0 |S^{(j)}| \leq M$ for some $M > 0$ and all $j = 1, \ldots, m$.*

Test statistics satisfying this assumption shall be referred to as shared randomness (or public coin) protocols.

The theorem below establishes the optimal rate when coordination through shared randomness is allowed. When the number of trials is small compared to the dimension (i.e. $m \lesssim d/\log m$), there is no difference between protocols that coordinate using shared randomness or those without coordination. In fact, the optimal rate ($\rho^2 \asymp \sqrt{d}/(\sqrt{m}n)$) in this case is reached by the test (7) or the ones below it, which do not employ shared randomness. However, when the number of trials is large compared to the dimension (i.e. $m \gtrsim d$), the testing rate substantially improves in the shared randomness protocols.

**Theorem 3.** *Let $S^{(1)}, \ldots, S^{(m)}$, $C_m$ and $T_\alpha$ satisfy Assumptions 2–4. Then there exists a constant $c > 0$ depending only on $L$, $p, q$ and $M$, such that if*

$$\rho^2 \leq c \frac{\left( \sqrt{m} \wedge \sqrt{\frac{d}{\log(m)}} \right) \sqrt{d}}{mn}, \tag{11}$$

*it holds that* $\sup\limits_{f \in H_\rho} \mathbb{P}_f (T_\alpha = 0) > 2/3$ *for all* $n, m, d \in \mathbb{N}$ *and any level* $\alpha \in (0, 0.1]$.

*At the same time, for all* $\alpha, \beta \in (0, 1)$ *there exists a constant* $C_{\alpha,\beta} > 0$ *depending only on* $\beta$, $L$, $p$, $q$, *the function* $\alpha \mapsto \kappa_\alpha$ *and* $M$, *such that if*

$$\rho^2 \geq C_{\alpha,\beta} \frac{\left(\sqrt{m} \wedge \sqrt{d}\right)\sqrt{d}}{mn} \tag{12}$$

*it holds that* $\sup\limits_{f \in H_\rho} \mathbb{P}_f (T_\alpha = 0) \leq \beta$ *for some test* $T_\alpha$ *of level* $\alpha$ *satisfying Assumptions 2–4.*

*Remark* 2. Similarly to Theorem 1 the choice of ranges $0 < \alpha \leq 0.1$ and $\beta = 2/3$ in the lower bound result is arbitrary, other choices are also possible as presented in the proof.

A shared randomness method that attains the rate in (12) is given next. Consider drawing an orthonormal $d \times d$ matrix $U$ taking values from the uniform measure on such matrices. As a test statistic, each trial computes $(\sqrt{n}UX^{(j)})_1$, which is a $N(0, 1)$ random variable under the null hypothesis. A level $\alpha \in (0, 1)$ meta-level test is then given by combining the local test statistics as

$$T_\alpha := \mathbb{1}\left\{ \Big| \frac{1}{\sqrt{m}} \sum_{j=1}^{m} (\sqrt{n}UX^{(j)})_1 \Big| \geq \Phi^{-1}(1 - \alpha/2) \right\}, \tag{13}$$

where $\Phi$ is the standard Gaussian CDF. The core idea here is that for each trial, the same 1-dimensional projection of the $d$-dimensional data is computed, where the projection is taken uniformly at random and the test is conducted along the projected direction. The above method corresponds to Stouffer's method for the p-values $p^{(j)} = \Phi(\sqrt{n}(UX^{(j)})_1)$ for $j = 1, \ldots, m$. Lemma 8 in the appendix shows that the above test attains a small Type II error probability whenever $\rho^2 \gtrsim d/(mn)$.

## 3 Examples for meta-analysis methods

Combinations of independent test statistics that fall into the framework of Assumptions 1– 4 are subject to the rate optimality theory established by the main theorems in Section 2. In this section, we look into common methods for combining p-values, e-values and other test-statistics, as mentioned in the introduction.

When the distribution under the null hypothesis of the test statistics are known, certain combinations are natural. For example, the sum of normal or chi-square test statistics is again normal or chi-square distributed, respectively. Similarly, voting based mechanisms typically rely on summing Bernoulli random variables. It is easy to see that these and similar combinations methods fall into the framework of Assumptions 1–4.

For more specific test statistics, such as p-values or e-values, many general combination methods have been introduced in the literature. We cover some of the most prominent combination approaches for p-values and e-values in Section 3.1 and Section 3.2, respectively. The list of methods is certainly non-exhaustive and many more combination methods exist, but they serve as context for the range of techniques covered by our general theory. Our main results allow establishing lower bound rates for the ones listed below, whilst in Sections 2.1 and 2.2 attainability of these rates by some of the listed methods was exhibited.

### 3.1 Combinations of p-values

If $p^{(1)}, \ldots, p^{(m)}$ are p-values obtained from $m$ independent test statistics concerning the same hypothesis, then under the null $p^{(j)} \sim^{iid} U(0, 1)$. One can aim to combine the $m$ p-values to form a test $T_\alpha \equiv T_\alpha(p^{(1)}, \ldots, p^{(m)})$ with Type I error probability $\alpha$, which hopefully has higher power than a test based on one of the individual p-values. Below we list standard methods in the literature.

- Fisher's method [18]. Because the variables $-2\log p^{(j)}$'s are iid $\chi_2^2$-distributed under the null hypothesis, their sum follows a $\chi_{2m}^2$-distribution. Therefore the combination method $\sum_{j=1}^{m} -2\log p^{(j)}$ results in a $\chi_{2m}^2$ distributed random variable, and the corresponding quantile function provides level-$\alpha$ one-sided tests at the meta-level.

- Similar flavour to Fisher's method are the combinations $\sum_{j=1}^{m} -\log(1-p^{(j)})$ (Pearson's method [32]), $\sum_{j=1}^{m} -\log p^{(j)}(1-p^{(j)})$ (the logit method / Mudholkar and George method [31]) and $m^{-1/2}\sum_{j=1}^{m}(p^{(j)} - 1/2)$ (Edgington's method [14]).

- Order-based methods such as Tippett's method [42] based on $\min\{p^{(1)}, \ldots, p^{(m)}\} \overset{H_0}{\sim}$ Beta$(1, m)$.

- Methods based on inverse CDF's, such as by Stouffer et. al [36] based on $m^{-1/2}\sum_{j=1}^{m}\Phi^{-1}(p^{(j)}) \sim N(0, 1)$ under the null hypothesis.

- Generalized averages as considered in [48], $T_\alpha = \mathbb{1}\left\{a_{r,m}M_{r,m}(p^{(1)}, \ldots, p^{(m)}) \leq \alpha\right\}$, where $M_{r,m}(p^{(1)}, \ldots, p^{(m)})$ equals $\left(m^{-1}\sum_{j=1}^{m}(p^{(j)})^r\right)^{1/r}$ for $r \in \mathbb{R} \setminus \{0\}$, the geometric mean, minimimum (i.e. Tippett's method) and maximum for $r = 0$, $r \to -\infty$, and $r \to \infty$, respectively. For $r \in \{-\infty\} \cup [1/(m-1), \infty]$, $a_{r,m}$ can be taken to obtain precisely level $\alpha$ tests (i.e. $\mathbb{P}_0 T_\alpha = \alpha$). We note that this means that canonical multiple testing methods (see e.g. [19]) such as Bonferroni's correction (which corresponds with taking as $M_{r,m}$ the minimum and $a_{r,m} = m$) also fall within our framework.

Lemma 1 below shows that all the methods mentioned above fall into the framework of Assumptions 1–4. This means that the error rate lower bounds of Theorem 1 and Theorem 3 respectively, apply to the p-value combination methods listed above. That is, one cannot attain a better separation rate when considering the worst case Type II error probability for the alternative hypothesis in (2), with any of the p-value combination methods listed above. Whether Assumption 1 or 4 applies depends on whether shared randomness is used in generating the p-values. To confirm that Assumptions 3 and 2 apply to tests based on the combined p-values, some algebra is needed. The proof of the lemma is deferred to the appendix.

**Lemma 1.** *Consider p-values $p^{(1)}, \ldots, p^{(m)}$, where each $p^{(j)}$ depends on the local data $X^{(j)}$ and possibly local randomness that is independent of the data. For each of the combination methods for p-values mentioned above and corresponding test $T_\alpha$ of level $\alpha \in (0, 1)$, the conclusions of Theorem 1 holds.*

We remark that the p-values are obtained using shared randomness (i.e. in the sense of Assumption 1), the lower bound rate of Theorem 3 applies. Furthermore, as exhibited in Sections 2.1 and 2.2, for p-values corresponding to well chosen test statistics, these combination methods can achieve the theoretical limits established in Theorems 1 and 3, respectively.

### 3.2 Combining e-values

An *e-value* is a nonnegative random variable $E$ such that $\sup_{\mathbb{P}_0 \in H_0} \mathbb{P}_0 E \leq 1$. The *threshold test corresponding to $E$ of level $\alpha$* is $\mathbb{1}\{E \geq \alpha^{-1}\}$. This test yields a so called strict p-value; for $\mathbb{P}_0 \in H_0$ we have $\mathbb{P}_0(E \geq \alpha^{-1}) \leq \alpha$ by Markov's inequality.

E-values lend themselves for combining outcomes of independent studies for two main reasons. First, they are easy to combine, see Section 4 in [49] for an indepth discussion of specific combination functions for independent e-values. Second, they are robust to misspecification and offer optional stopping/continuation guarantees [22]. Common examples of e-values are Bayes factors and likelihood ratios, which are nonnegative and have expectation equal to 1 in the case of a simple null hypothesis such as considered in this article.

Several combination methods (e-merging functions) were proposed in the literature. For instance, the product of independent e-values is also again an e-value. This was shown to weakly dominate any other combination of independent e-values in the sense that $\Pi_{j=1}^{m}E^{(j)} \geq C_m(E)$, for any $E = (E^{(j)}) \in [1, \infty)^m$ and $E \mapsto C_m(E)$ such that $C_m(E^{(1)}, \ldots, E^{(m)})$ is an e-value for any independent e-values $E^{(1)}, \ldots, E^{(m)}$, see [49]. Similarly, the average of e-values is again an e-value. The product and the average are *admissible* in the sense that there is no e-merging function that strictly dominates them on $[0, \infty]^m$. The lemma below shows that these two, arguably most prominent e-value combination methods fulfill Assumptions 1– 4 and hence the lower bounds derived in Theorems 1 and 3 apply.

**Lemma 2.** *Consider e-values $E^{(1)}, \ldots, E^{(m)}$, where each $E^{(j)}$ depends on the local data $X^{(j)}$ and possibly local randomness that is independent of the data. Let $C_m : \mathbb{R}^m \to \mathbb{R}$ correspond to either the average or the product and let $T_\alpha$ be the corresponding threshold test of level $\alpha \in (0, 1)$,*

$$T_\alpha = \mathbb{1} \left\{ C_m(E^{(1)}, \ldots, E^{(m)}) \geq \alpha^{-1} \right\}.$$

*If $C_m$ is the product, assume in addition that $\mathbb{E}_0 |\log E^{(j)}|$ is uniformly bounded. Then, the conclusion of Theorem 1 holds. In case the e-values are generated using shared randomness, then Theorem 3 applies.*

## 4 Simulations

In this section, we investigate the numerical performance of the testing strategies outlined in Section 2.1 on synthetic data sets. We compare the tests based on their receiver operating characteristic (ROC) curve. For a range of significance levels we compute for each tests the "true positive rate" (TPR) and "false positive rates" or (FPR), i.e. the fraction of the simulation runs in which the test correctly identifies the underlying signal, falsely rejects the null hypothesis, respectively. Plotting the TPR against the FPR (both given as a function of the significance level) provides us the ROC curve, visualizing the diagnostic ability of the test.

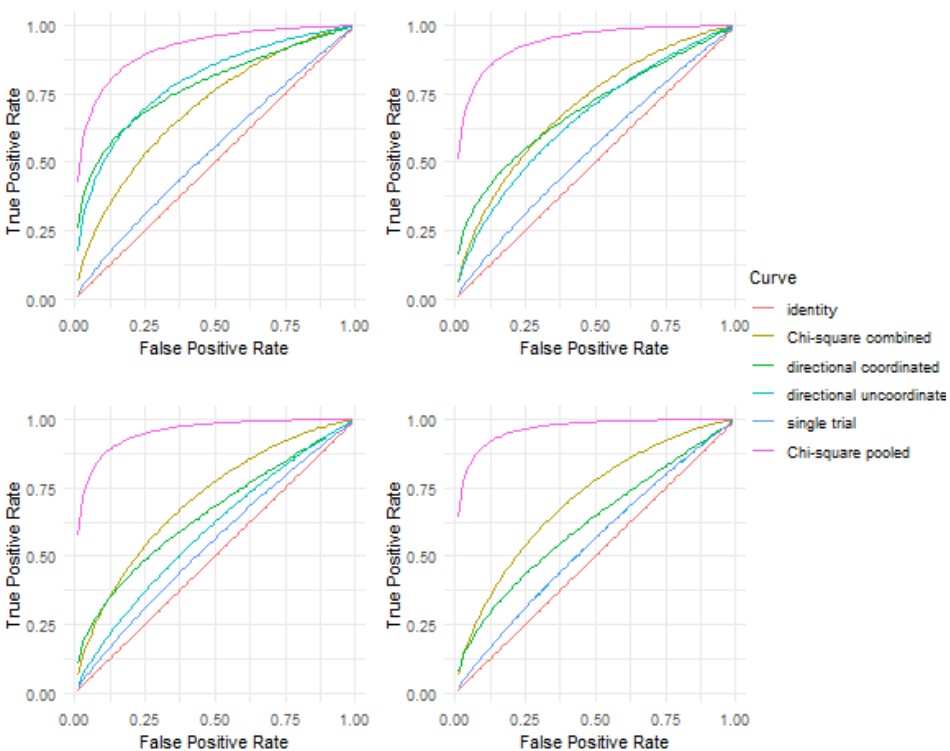

Figure 1: ROC curves for different values of $d$, whilst keeping $m = 20$, $n = 30$, $\rho^2 = \sqrt{d}/(4n)$. From left to right, top to bottom: $d = 2$, $d = 5$, $d = 10$, $d = 20$.

In our simulations we set $m = 20$, $n = 30$, let $d$ range from 2 to 20 and take $\rho^2 = \sqrt{d}/(4n)$. This value of $\rho^2$ corresponds to a signal that is almost indistinguishable from noise using just a single trial, whilst consistently detectable if the data were to be pooled with $m \approx 20$ (which increases the signal size to noise ratio effectively by a factor $\sqrt{20} > 4$. For each level $\alpha \in \{0.01, 0.02, \ldots, 0.99\}$ we compute the power for different combination strategies 100 times, each time drawing a different $f \in \mathbb{R}^d$ with $\|f\|_2 = \rho$ according to $f_i = d^{-1/2}\rho R_i$ and $R_i$ iid Rademacher random variables for $i = 1, \ldots, d$. As combination strategies, we compare the strategies (7), (13) and (8) from Section 2.

which are called "chi-square combined", "coordinated directional" and "uncoordinated directional" in the legend of Figure 4. In addition, we display the ROC curves for the chi-square test based on pooled data ("chi-square pooled") and that of a single trial ("single trial").

We make the following observations, in line with our theoretical findings. The meta-analysis methods based on combining the locally optimal chi-squared test statistics (yellow curves) substantially outperformed the chi-squared test statistics based on a single trial (blue curve), but was substantially worse than the chi-square test based on the pooled data (pink curve). Second note that the large dimensional case ($d = 10$ and $d = 20$) the best strategy is indeed to combine the local chi-square statistics (yellow curve), while in the low dimensional setting ($d = 2$) it is more advantageous to combine the directional test statistics $X_i^{(j)}$ (blue curve). Finally, note that allowing coordination between the trials by a shared randomness protocol can result in improved performance (green curve) compared to the independent experiments (blue curve). In fact this approach provides the best meta-analysis method in the small dimensional setting (e.g. $d = 2$ and $d = 5$ for small $\alpha$, which is the most interesting case).

In the appendix, Section A.6, we explore eight additional simulation settings, where we consider larger values of $d$ and $m$. Whilst these simulations do not reveal additional phenomena to the ones observed in Figure 4, they do give insight into the relative performance of the testing methods for different values of $d$ and $m$.

## 5   Discussion

We briefly summarize our main contributions and discuss possible extensions and research directions. First, by establishing a connection between meta-analysis and distributed learning under communication constraints, we have provided a unified, theoretical framework for evaluating the behaviour of standard meta-analysis techniques. In our analysis, we considered the many normal means model, but these results can be extended to other more complex models as well, building on the connection with distributed computation. For example, minimax estimation rates under communication constraints were derived for other parametric models [53], density estimation [8], signal-in-Gaussian-white-noise [54, 38, 10], nonparametric regression [37] and in abstract settings [52] including binary and Poisson regression, drift estimation, and more. The normal means model allows for a tractable analysis, but results in this model are known to extend to more complicated models, such as discrete density testing (see e.g. [11]). With the due technical work, our results are expected to translate to these settings as well, but we leave this for future endeavor.

In the normal means model we show that by combining the locally optimal chi-square statistics at a meta-level one can gain a factor of $\sqrt{m}$ compared to using a single trial. Nevertheless, regardless of the choice of the combination method, a factor of $\sqrt{m}$ is lost compared to the scenario when all data from all trials are at our disposal. This loss is clearly visible even in small sample sizes, dimensions and trial numbers, as demonstrated in our numerical analysis, as can be seen in the corresponding ROC curves. For more complex models, such a numerical study can be a first step to quantify the efficiency of the meta-analysis method. We have also shown that in the small dimension - large number of trials setting combining the locally optimal chi-square statistics (or any rotationally invariant statistics for that matter) results in information loss and sub-optimal accuracy. In this case, better rates can be attained by test statistics based on the direction of the observations combined at the meta-level. In practice, one often cannot choose which test statistics can be obtained from independent trials. In such cases, the $\sqrt{m}$-factor loss in the case of e.g. rotationally invariant test statistics is of interest when considering power calculations. Meta-analysis approaches based on directional test statistics are designed for scenarios where individual datasets are not centrally collected, but there is some level of coordination among experimenters.

The assumption throughout the paper of homogeneity between the trials (i.e. each trial consisting of the same number of observations) simplifies the presentation, but the results can be extended to cases where the number of observations in each trial differ by constant factors. Situations where the number of observations differs greatly (e.g. $k \ll m$ trials have as much observations as the other $m - k$ trials combined) are certainly of interest, but beyond the scope of this paper.

**Acknowledgements:** Co-funded by the European Union (ERC, BigBayesUQ, project number: 101041064). Views and opinions expressed are however those of the author(s) only and do not necessarily reflect those of the European Union or the European Research Council. Neither the European Union nor the granting authority can be held responsible for them. This research was partially funded by a Spinoza grant of the Dutch Research Council (NWO).

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
