# OpenReview forum: "Optimal testing using combined test statistics across independent studies"
_NeurIPS.cc/2023/Conference — NeurIPS 2023 poster_

### Official Review · Reviewer_ThPF · 2023-06-29

**Soundness:** 4 excellent
**Presentation:** 4 excellent
**Contribution:** 2 fair
**Rating:** 6
**Confidence:** 3

**Summary:**

[Update1: During the rebuttal, I updated my score from 5 to 6. The reason is that I want to stronger weigh in that the paper is technically solid. My concern whether NeurIPS is a perfect fit for this paper remains, and I ask the AC to judge that part.]
The paper studies aggregation strategies for combining test statistics obtained from independent experiments. It is intuitively clear that the optimal strategy would be to aggregate the whole *data* and then compute one test statistic on the entirety of the data. However, the authors make the assumption that from each experiment one can only use a single real number (aka test statistic) for the final analysis.

The setting they consider is that they have observations of random variables

$$
X^{(j)} = f + \frac{1}{\sqrt{n}} Z^{(j)},
$$
where $Z^{(j)}$ are independent $d$ dimensional normals and $n\in \mathbb{N}$ and additional parameter. The null hypothesis is $f=0$ which is tested against $\|f\|_2 \geq \rho$ for some $\rho >0$. If pooling the data was allowed, the chi-square test is optimal for this setting.

They derive theoretical results that give two different regimes:

- When $m \lesssim  d^2$ the optimal rate can be achieved by combining the individually optimal test statistics $\|\sqrt(n) X^{(j)}\|_2^2$, which is undirected.

- When $m \gtrsim d^2$ then taking the directionality into account leads to better rates. Two examples they discuss are (phrasing in my own words that the authors might adapt):
  - Split the observations into $d$ groups of size $\sim m/d$ and test with the $i$-th group whether the null hypothesis holds for the $i$-th dimension. Then combine those.
  - Choose a 1-dimensional projection of the $d$-dimensional data uniformly at random. Aggregate all observations along this projections (now these are 1-dimensional values, as required in the setting) and combine those directly to test the null hypothesis along the projected direction. (Note that this requires shared randomness between all experiments).

Finally the authors demonstrate their theoretical findings with toy experiments and numerically confirm their findings.

Overall I enjoyed reading the paper and was able to learn something new. Thank you to the authors.


**Strengths:**

- The paper is extremely well written and has "textbook" quality. It focuses on a simple toy problem and provides an exhaustive analysis thereof.
- Combining outcomes from different experiments is certainly an important problem in statistics.
- The paper seems original, however, admittedly I am unable to judge to what extent similar analyses exist in the stats literature.
- The theory and experiments are in perfect accordance and complement each other.
- The paper provides a good overview of existing methods to combine test statistics and discusses both $p$- and $e$-values as examples. They relate existing methods to their contribution.

**Weaknesses:**

1. My first point concerns the significance of the approach. In the introductions the paper states
  > Given multiple data sets relating to the same hypothesis, one would like to combine the evidence.
 Sometimes, the full data sets are not available (e.g. due to privacy or proprietary reasons) or difficult
 to combine directly (e.g. due to the different experimental or observational setups). In such cases,
 the analysis must be carried out on the basis of the published results for each of the studies.

    - This motivation reads that each study publishes a test statistic without knowing of the others. I thus think that the second type of approaches the authors provide does not fall in this category.
    - The second type of approaches require that the "meta-analyser" can make some choices of what statistic the individual studies report. Hence *somewhere* all data must still be available. It is unclear why we cannot use that then. I think this needs more motivation.
    - I presume that the first approach of combining the undirected tests has been studied exhaustively.

2. While the paper is relevant for machine learning in general, it is in itself a very statistical paper. No learning involved etc. Hence, I am posing the question whether NeurIPS is the appropriate venue. I would see a stats journal as a better fit.

3. I think the paper could be a bit more accessible if a bit more intuition about the approaches is provided. I wrote my understanding in the summary, maybe the authors want to correct/modify this and include something like that.

**Questions:**

Please comment my first two concerns above in the rebuttal. Note that my reservation against acceptance is based on those comments. And I am happy to increase my score after the rebuttal if I am convinced otherwise.


Minor:
- l 124 "where" --> "were"
- l. 339 "form" --> "from"

**Limitations:**

This paper only studies a simple toy problem.

Limitation of my review: I did not read the appendix.

---

> ### Author Rebuttal · Authors · 2023-08-09
>
> We appreciate the time and effort the Reviewer has put in evaluating our work. The Reviewer also raises a few concerns to which we respond below.
>
>
> *1 -- Significance of the approach:*
>
> We agree that in case of independent trials typically the test statistics are set independently from the number of trials. In such cases, the best attainable rate is $\frac{\sqrt{d}}{\sqrt{m}n}$. This is still novel and an important contribution however; theoretically capturing explicitly the loss in terms of error rate (i.e. the $\sqrt{m}$ worse error rate) when conducting meta-analysis in such scenarios. It is important for practitioners conducting meta-analysis to be aware of this potential loss in statistical power. In our revised manuscript we have highlighted these more prominently and provided an extended description of Theorem 4, emphasising this point.
>
> The second meta-analysis approach based on directional test statistics is designed for scenarios where individual datasets are not centrally collected, but there's coordination among experimenters. Even if one can make some choices regarding the statistics, it does not mean that all raw data must be accessible. The result shows that (when designing experiments) advance coordination can be beneficial (e.g. one could consider reporting a directional statistic, employ shared randomness with a public seed, etc.).
>
> We have revised our explanation in the article to better reflect these contributions.
>
> *2 -- The connection to learning theory:* We agree with the reviewer that our paper could be published in a (good) statistics journal as well. However, despite the statistical nature of the paper, we believe that the problem we investigate is relevant to various fields in machine learning, particularly in scenarios where data/evidence aggregation occurs. So far these areas had limited theoretical underpinning and deriving guarantees (from a statistical perspective) helps better understanding these phenomena and improves the explainability of these methods. We believe that NeurIPS provides a suitable multidisciplinary platform to reach an audience interested in these intersections between statistics and machine learning and also appreciate a theoretical oriented work.
>
> *2 -- Accessibility:*
> We will follow your advice to include a section that translates the technical details into an intuitive understanding. We thank the Reviewer for their take on the regimes and intuition for which testing strategies are optimal, we have adopted this intuition in the introduction and setting sections of our new version of our article. This should help readers unfamiliar with the depth of the statistical methods to grasp the key concepts.
>
> We thank the Reviewer for pointing out several of the typos, these are corrected in the new version of the paper.

---

> > ### Comment · Reviewer_ThPF · 2023-08-10
> > **Thanks for the rebuttal**
> >
> > I thank the authors for their reply. I do not have any further questions.
> >
> > I  encourage the authors to careful write out their distinction between a pure meta-analysis of, say, p-values and an analysis that allows for coordination between the trials. While the first one is clear, I believe the latter needs more motivation.
> >
> > Nevertheless after reading the other reviews and the rebuttals, **I will increase my score from 5 to 6.** I think the paper is a technically very solid contribution.
> >
> >
> > [Note: ~It seems that currently I am unable to adjust my score. But I will do so later.~ Review is now updated]

---

> > > ### Author Response · Authors · 2023-08-10
> > > **reply**
> > >
> > > We thank the Referee for the reply and the suggestion, we will follow it in the revision. We also appreciate that the Reviewer went through all the comments and rebuttals and we thank her/him for the additional point.

---

### Official Review · Reviewer_GjrH · 2023-07-02

**Soundness:** 3 good
**Presentation:** 2 fair
**Contribution:** 2 fair
**Rating:** 5
**Confidence:** 3

**Summary:**

This theory paper provides a minimax lower and upper bounds for the testing risk (sum of Type-I and Type-II errors) for different combination methods in the specific setting of many normal means model. With the testing goal is to detect the presence or absence of the signal component in this normal means model, their results show several combination methods of test statistics (e.g. aggregate p-values and e-values) cannot consitently detect signals below a certain threshold that depend on the number of trials, samples and dimensions of the problem.

**Strengths:**

* The theoretical results are sound and based on several established techniques in distributed testing, assuming several assumptions  hold true. These are proof to be indeed the case in the Appendix for the test statistics combination methods proposed in Section 3 of the main text.
* The paper is mostly well-written.

**Weaknesses:**

1. In general I think the phrasing of on the paper's contributions could be make clearer in the last part of the introduction section.
2. The authors should discuss more on the relevance of the setting -- the many normal means model -- in some more concrete applications. It is arguable that although this is a theoretical paper, the theories inside it is an attempt to quantify a very practical problem of meta learning. I see a lack of evidence for the popularity of many normal means in practice.
3. Slightly related but not as equally importance, but the authors should have acknowledged that a limitation of their work is that the theoretical results only hold with many normal means model assumption.
4. Experimental results could include more settings with a variety number of of sizes/dimensions (perhaps in the appendix) to support the theory.

**Questions:**

* I have only skimmed through the proof in the appendix, so this might be my mistake, but I do not see the appearance of the $\epsilon$ term for binary approximation of the statistics in the main results. Could the authors clarify on this point?

**Limitations:**

* See weaknesses.

---

> ### Author Rebuttal · Authors · 2023-08-09
>
> We thank the Reviewer for reviewing our manuscript and the constructive feedback. We address below the raised concerns point-by-point.
>
> *1 -- Phrasing of on the paper's contributions:* Based on your feedback, to futher improve our mansucript, we have restructured and emphasised the contributions in the last part of the introduction to clearly outline the scope of our paper and our findings.
>
> *2 -- Relevance of the setting:* The many normal means model is indeed a specific statistical model, but it is foundational to statistical theory, capturing phenomena which readily extend to more complicated statistical models, for instance nonparametric regression, classification, density estimation, just to name a few. In the revised manuscript we provide a more extensive description this model and its connection to other, practically more important statistical models, citing literature that further expand on this connection:
>
> * Large-scale inference: empirical Bayes methods for estimation, testing, and prediction - Efron (2012)
> * Introduction to Nonparametric Estimation - Tsybakov (2003)
> * Mathematical foundations of infinite-dimensional statistical models -  Giné and Nickl (2015)
> * Deficiency distance between multinomial and multivariate normal experiments - Carter (2002)
>
> It is because of this strong connection to other statistical models that the many-normal-means model serves as a benchmark to derive error rates, even when one has more complicated statistical settings in mind, as often the error rates extend. See for example:
> * Information-theoretic lower bounds for distributed statistical estimation with communication constraints (https://proceedings.neurips.cc/paper_files/paper/2013/file/d6ef5f7fa914c19931a55bb262ec879c-Paper.pdf) - Zhang, Duchi, Jordan and Wainwright (2013)
> * Handling Sparsity via the Horseshoe (https://proceedings.mlr.press/v5/carvalho09a/carvalho09a.pdf) - Carvalho, Polson and Scott
> * Needles and straw in haystacks: empirical bayes estimates of possibly sparse sequences (https://arxiv.org/pdf/math/0410088.pdf) -  Johstone and Silverman (2004)
>
> *3 -- Acknowledgement of limitations:* We have further emphasised in the revised version that our results concern the many-normal means model. Extension to more other, more complex models (see examples above), are left for future work.
>
> *4 -- Experimental:* We have extended the simulation settings, by expanding the appendix. The simulations now include a variety of high-dimensional settings and differing sizes.
>
> *Question concerning $\epsilon$ dependence:* The $\epsilon$ term is absorbed in the constant $c > 0$ in the main results to improve the readability. If explicit dependence on $\epsilon$ is desired in the main results, we will gladly change this in the new version.
>
> In conclusion, based on your feedback, we have identified areas where we enhanced our manuscript, we hope that the revisions will address your concerns.

---

> > ### Comment · Reviewer_GjrH · 2023-08-11
> > **Thank you for your rebuttal**
> >
> > I think the author have answered all my questions thoroughly. However, I agree with one of the reviewer point that in general this work leans on more of a statistical methodology paper, therefore I still maintain my score as Weak Accept, as I do not see a major problem with it.

---

> > > ### Author Response · Authors · 2023-08-12
> > > **reply**
> > >
> > > We are grateful to the Reviewer for their response and their thorough examination of all the comments and rebuttals. For the purpose of clarification, we would like to ask whether the final score of the reviewer is "Weak Accept" (6) or a "Borderline Accept" (5)? Thank you in advance for your reply.

---

> > > > ### Comment · Reviewer_GjrH · 2023-08-14
> > > > **Sorry for the confusion**
> > > >
> > > > What I meant is Borderline Accept (5) score, but of course this reflects my opinion that I would not be upset if the work is accepted to NeuRIPS, as I saw some clear contributions to the conference.

---

### Official Review · Reviewer_vLRc · 2023-07-03

**Soundness:** 3 good
**Presentation:** 4 excellent
**Contribution:** 3 good
**Rating:** 7
**Confidence:** 2

**Summary:**

Authors study a problem of optimal combination of p-values in a meta-analysis context. Specifically, they focused on characterizing the minimax separation rate for a family of "smooth-ish" combination methods that aggregate p-values (or e-values). They show that:
* The family contains a lot of methods that are used in practice.
* The separation rate for methods in the family is nearly optimal.
* Optimal combination method depends on the problem setting (sample size, number of p-vals, dimension of the problem); they describe practical consequences.

Furthermore, they explored how the rate can be improved by allowing coordination between the experiments that generate the p-values.

**Strengths:**

The paper is written in a clear and without excessive statistical jargon. For instance, the concept of separation rate is introduced and explained in a simple and intuitive way, allowing readers to understand the results presented in the paper (in contrast to the mat description found in "Non-asymptotic minimax rates of testing in signal detection"). The significance of the results is solidly established on two grounds: derivation of a separation rate and practical guidance for method selection based on n, m, and d (Section 2.2 and 2.3 add extra value).  From my (limited) understanding of the literature, these results are both novel and original.

**Weaknesses:**

I am genuinely surprised that this problem has not been previously studied. While preparing the review, I came across similar/related results concerning the Family-Wise Error Rate (FWER) in the paper "Family-Wise Separation Rates for multiple testing." However, the combination methods studied are different (Holm–Bonferroni procedure type).

I'd suggest a more comprehensive and thorough discussion of the existing literature on minmax testing for multiple hypothesis testing is added to the paper.

**Questions:**

See above

**Limitations:**

yes

---

> ### Author Rebuttal · Authors · 2023-08-09
>
> We sincerely appreciate the time and effort spent on evaluating our paper, the positive feedback and the insightful suggestions. Below, we address the specific suggestions raised by the Reviewer.
>
> *Novelty of the results:* Yes, indeed, we were also surprised by the lack of theory for meta-analysis in hypothesis testing with a composite alternative hypothesis. Actually, this was the main motivation of our research, to fill this gap in the literature.
>
> *Connection to multiple testing:* We agree with the reviewer that multiple testing has lot of similarities to the concerned meta analysis problem. The common goal is to aggregate somehow the outcome of multiple (often independently executed) tests. However, there are also major differences. In the meta-analysis  framework (e.g. when combining p-values) one considers a given hypothesis and executes multiple experiments on deciding its validity. Then the outcomes of these tests are combined for a more accurate decision about the common hypothesis of interest. In contrast to this in multiple testing in each experiments (possibly) different hypotheses are considered and the goal is to decide which ones should be rejected. Here the main challenge is to obtain uniform guarantees over all hypotheses (e.g. by controlling the FWER or FDR). Following the suggestion of the Referee we discuss then  connection of these related but different testing problems.

---

> > ### Comment · Reviewer_vLRc · 2023-08-15
> >
> > I'm confused by this comment. Bonferroni would do just fine achieving a goal of combining p-values, Bonferroni combination function is not smooth and does not fall into your framework.
> >
> > Your answer does not  give me great confidence that you actually reviews multiple hypothesis testing thoroughly.  Please formalize the difference between multiple testing and meta analysis problem.
> >
> > Temporarily moving down to 5.

---

> > > ### Author Response · Authors · 2023-08-16
> > >
> > > We are sorry to hear that our response caused confusion. We provide below a detailed response on the difference between multiple testing and meta-analysis, in addition to explaining that our framework actually includes Bonferonni's method.
> > >
> > > **On Bonferonni's method:** Bonferonni's method would entail combining p-values as $m \cdot \min \{p^{(1)},\dots,p^{(m)} \}$ (see e.g. display (1) in [VOVK, V., AND WANG, R. Combining p-values via averaging - Biometrika 107]). This is in fact covered by our framework, on page 7 we discuss generalized averages as defined for example in [VOVK, V., AND WANG, R. Combining p-values via averaging - Biometrika 107]. This framework also encompasses the Bonferonni correction, which is a generalized average (with $a_{r,m} = m$ in the notation of page 7 of our paper). We would also like to highlight that our framework is more general than just smooth combination functions, as Assumption 3 concerns only Hölder continuity (which is satisfied for Tippett's method or a Bonferonni correction as well).
> > >
> > > We did not highlight this method as it is more conservative than e.g. Tippett's method ($ \min \{p^{(1)},\dots,p^{(m)} \}$), and is often considered when the p-values combined might have dependencies because they e.g. concern the same data, which is the case in multiple testing, see the definition below.
> > >
> > > **Comparing the mulitple testing to meta-analysis:**
> > >
> > > *Multiple testing*: Let $X$ be some data drawn from an unknown distribution $P$. Based on the true distribution $P$, a hypothesis $H$ is either true or false; which we denote by $H$ is being true if it belongs to a set $\mathcal{T}_0$ and false if it belongs to $\mathcal{T}_1$ otherwise. Given a collection of such hypotheses $\mathcal{H}$, using the data $X$ one tries do discern
> > > $$H \in \mathcal{T}_0 \text{ versus } H \in \mathcal{T}_1$$
> > > for all $H \in \mathcal{H}$ simultaneously. This very general definition of multiple testing, see for example [FROMONT ET AL. Family-Wise Separation Rates for multiple testing - Ann. Statist. 44(6)] Prototypical examples are testing for each gene in a sequence separately whether the gene plays a role in a given disease, or testing the returns of different portfolios, for finding which portfolios have higher than market returns.
> > >
> > > For example, a multiple testing problem in the context of the many-normal-means model considered in our paper is
> > > $$H_{0k}: f_k = 0 \text{ versus } H_{1k}: |f_k| \geq \rho_k$$
> > > for $k=1,\dots,k$, given data $X = f + \frac{1}{\sqrt{n}}Z$, $f \in \mathbb{R}^d$. For such a collection of hypotheses, one tries to discerns multiple, different hypotheses on the basis of the same data.
> > >
> > > Standard approaches in multiple testing are: Bonferonni correction and Holmes method (to control the family-wise error rate) and Benjamini-Hochberg (controlling the false discorvery rate).
> > >
> > > *Meta-analysis* can be performed when there are multiple scientific studies addressing the *same question* (see e.g. [Hedges et. al - Introduction to meta-analysis] or Wikipedia). In our analysis, we consider testing, where $m$ studies address the *same hypothesis* and the goal is to combine the study outcomes (e.g. their reported p-values). Prototypical examples would be multiple experiments conducted to establish whether a given drug has *any* effect (e.g. whether a given blood pressure medication indeed lowers the bloodpressure), or multiple studies concerning the question whether smoking causes cancer.
> > >
> > > In our setting, we only consider the same hypothesis (i.e. $\mathcal{H}$ is a singleton) in each study:
> > > $$H_{0}: f = 0 \text{ versus } H_{1}: \|f\| > \rho. $$
> > >
> > >  *In conclusion:* Although a Bonferonni correction falls within our framework, it is unnecessarily conservative as we do not use the same data for testing more than one hypothesis. Therefore we did not explicitly mention it, as e.g. Tippett's method is more appropriate for our setting. Nevertheless, in our updated version we explicitly refer to this method as well as an example of a generalized average. We hope that our definitions above highlighting the conceptual differences of meta-analysis and multiple testing are satisfactory.

---

> > > > ### Comment · Reviewer_vLRc · 2023-08-21
> > > >
> > > > Thank you for your response. It is satisfactory, and I am increasing the score. I appreciate the effort put into the rebuttal and the paper.

---

> > > > > ### Author Response · Authors · 2023-08-21
> > > > >
> > > > > We thank the Reviewer for the reconsideration and for the increased points.

---

### Official Review · Reviewer_MHU2 · 2023-07-24

**Soundness:** 3 good
**Presentation:** 3 good
**Contribution:** 2 fair
**Rating:** 5
**Confidence:** 2

**Summary:**

The paper addresses the problem of combining test statistics from multiple independent studies in the context of null-hypothesis significance testing. The authors derive a mathematical framework to quantify the cost of compressing multiple independent trials of a study into one real-valued test statistics, and they derive minimax lower and matching upper bounds for the testing errors. The many normal means models is used as toy example.

**Strengths:**

The paper addresses the problem of combining the results of multiple empirical studies towards one common hypothesis, an important problem in meta-analysis.

**Weaknesses:**

This submission seems to be out of the scope of NeurIPS. While the authors draw a connection between meta-analysis as used in statistics and meta-learning as used in the context of machine learning, this connection is not clarified further. For the rest of the paper, the authors seem to focus on the problem of meta-analysis.

It was challenging to read to paper as it lacks clarity in the introduction. To improve clarity, I suggest to start the introduction by clearly stating the problem that will be addressed in the paper and clearly introducing the terms used in the text. For example, in line 23, the authors introduce “meta-analysis”, (probably) referring to the technique of combining the results of multiple scientific empirical studies and then set it equal to “meta-learning”, referring to the machine learning technique of improving a learning algorithm to perform well multiple tasks.

I did not go through the technical parts of the paper in all detail. However, there are several statements in the paper that appear to be misleading, e.g.,

- line 68: “… includes many standard meta-learning techniques, for instance the standard p-value combination methods […]”. I am not aware of p-value combination being a standard technique in meta-learning.
- line 313: “Common examples of e-values are Bayes factors and likelihood ratios.” e-values and Bayes factors are closely related, but Bayes factors or likelihood ratios are not e-values, see https://arxiv.org/abs/1912.06116 Appendix A for clarifications.

**Questions:**

Can you clarify to connection between "meta-analysis" and "meta-learning" (meaning the approach reviewed in your reference number [14]?

**Limitations:**

The authors do not discuss limitations or potential negative impact of their work. Although being a more technical paper, I think it would have been appropriate to discuss the limitations of the meta-analysis approach as a whole.

---

> ### Author Rebuttal · Authors · 2023-08-09
>
> We thank the Reviewer for the effort of evaluating our paper and we are happy to hear that the Reviewer
>  shares the opinion that the problem is important to study. The Reviewer does not specifically comment on the soundness
>  of the mathematical framework, but highlights potential misleading use of terminologies and a lack of clarity in the
>  introduction, both of which we address below.
>
> *Connection to ML:*
> We have taken the advice of the Reviewer to heart and improved the clarity of the introduction by providing an
>  expanded and more clear explanation of the problem. We have also expanded the introduction to more explicitly describe
>  the connection and relevance to more modern machine learning applications.
>
> *Meta learning vs meta analysis:*
> In our analysis we used meta-learning / meta-analysis techniques as statistical and/or machine learning techniques distilling information across studies and different
>  data sets to form a more informed inference. We are sorry that using these terms basically interchangeably in our manuscript caused confusion. In the literature we did not find a standard interpretation of these terms (as is also discussed e.g. in the corresponding wikipedia page) and found that it depends on the specific community how they are interpreted.  However, to avoid confusion, we have decided to abstain from using the term meta-learning and stick to meta-analysis throughout the text.
>
> *Likelihood ratio as e-value:*
> Whilst it is true that in situations where the null hypothesis is composite, a Bayes factor or likelihood ratio is not necessarily an e-value, we are considering a setting in which the null hypothesis is simple. Any Bayes factor given by a likelihood ratio of
>  a mixture distribution $P_\pi := \int P_f d\pi(f)$ and the measure under the null hypothesis forms an e-value: $E := \frac{dP_\pi}{dP_0}$
>  satisifies $E \geq 0$ and $P_0 E = 1$. We have expanded on the statement: ``Bayes factors and likelihood ratios form common examples of
> e-values.'' to clarify that this should not be read in largest generality, but in the context of hypothesis tests such as considered in this article.
>
> We have added additional discussion concerning the limitations of the meta-analysis approach as a whole.
>
> We would like to thank the Reviewer again for their consideration.

---

> ### Comment · Reviewer_ThPF · 2023-08-15
> **Is Meta-Learning vs. Meta-Analysis clarified?**
>
> Dear reviewer colleague MHU2,
> in order to get a final judgment of the paper, I would like to understand whether your concerns have been addressed by the rebuttal or not?
> I share your definition of "Meta-Learning" and also think that at least within the NeurIPS community using "meta-analysis" is much more appropriate. But the authors promised to change that and I do not see any further problem with it. Do you?
>
> If it was clarified, then frankly I feel that a low score of 3 with a (quite high) confidence of 3 is not adequately reflected in your review. I think there is about a day left to ask further questions to the authors. I would like to be able to read their answer, if there are any questions still.
>
> Thank you!

---

### Official Review · Reviewer_qxf1 · 2023-07-25

**Soundness:** 3 good
**Presentation:** 3 good
**Contribution:** 3 good
**Rating:** 6
**Confidence:** 3

**Summary:**

The paper lies in the context of meta-analysis of multidimensional models. Usually, meta-analyses are performed by combining p-values or e-values. In both cases, the statistical power is not well-known. The authors provide a constrained framework of many means normal model. Based on this framework, they derive lower (theorem 1) and upper (theorem 2, for "rate optimal methods") bounds for testing methods. Theorem 3 takes advantage of possible shared randomness between trials, especially when the dimension is small relative to the number of trials considered. The authors then compare the performances of rate optimal methods (described in section 2.1), chi-square test on pooled data and single trial approach on simulated datasets.

**Strengths:**

The paper is well-written and organised. It provides a good overview of state-of-the-art combination techniques for meta-analyses and identifies the lack of knowledge about their relative power.
The paper derives bounds for the testing errors by introducing a principled mathematical framework based on multidimensional models, where a loss in power is expected. It also gives insights into rate optimal combination methods and the effects of sharing randomness between trials. The simulation study provides some results on comparing combinations methods for meta-analysis, which is not properly addressed in the current literature.
About the supplementary material: Proofs as well as an R script to reproduce the simulation study are provided.

**Weaknesses:**

The main weakness of this paper is the clarity of the mathematical developments. The framework implies several assumptions that could be explained more. The 13-page-long supplementary material provides proof of the theorems but is sometimes hard to follow.
In the theorems and their proofs, sometimes arbitrary values are chosen (for $\alpha$ and $\beta$ notably) but it seems to make the reasoning more confusing.

Also, when running the R script, the following error is returned:
Error: object 'dat_long' not found

**Questions:**

- $\mathbb{E}_0$ is first used in line 141 but not introduced beforehand. Is it the expectation under the null hypothesis?
- I understand that Assumption 1 aims at restricting the values of S. Is it possible to give a small interpretation of the assumption?
- Theorem 3 indicates that "there exists a constant $C_\beta$" but the formula indicates "$C_{\alpha,\beta}$". Is it a typo error?
- In theorems 1 and 3, arbitrary values are used for $\alpha$ and $\beta$, not in theorem 2. Why make the choice of using these values and not giving general results for the corresponding intervals of validity?
- A similar remark on the proofs, for example in proof A.1. Why use $\kappa_{1/10}$ and $\kappa_{1/8}$?
- It might also be more comprehensible to explicitly add the results taken from the literature and used in the proofs.
- The provided R script needs to be reviewed. When running it, I get the following error is returned:
"Error: object 'dat_long' not found"

I understand that the chosen mathematical framework provides lower and upper bounds for testing errors. The paper describes some meta-learning techniques that attain these bounds. I am not sure how the simulation study demonstrates the theoretical results. Is it by comparing these meta-learning techniques to the "chi-squared pooled" approach and the single trial approach?
The indicator of performance for "optimal testing" is the ROC curve. Would it be interesting to consider other criteria, such as sensitivity, specificity, precision, or F-score?

Overall, I encourage the authors to add more explanations and interpretations, especially in the mathematical development part. Note that the 9-page limit does not include references. It might also be worth submitting the paper to another journal where the format might be more adequate.


**Limitations:**

The authors have delimited the framework of their contribution by providing constraints inherent to their model.

---

> ### Author Rebuttal · Authors · 2023-08-09
>
> We thank the Reviewer for the thorough analysis and thoughtful feedback, we appreciate the time and effort. We address below the questions and comments of the Referee point-by-point:
>
> * Thank you for pointing this out, we have clarified in the new version of the paper that $\mathbb{E}_0$ corresponds to the expectation of $\mathbb{P}_0$, i.e. the probability distribution corresponding to the null hypothesis.
>
> * Assumption 1 requires that the combined test statistics can be "rescaled'' appropriately. This assumption works in conjunction with Assumption 2, as the degree of Hölder continuity of the combination function depends on the ``scale'' (or rather, the effective support) of the underlying test statistics. We provide a more detailed explanation of the conditions in the new version of the article.
> * The $\alpha$ subscript is indeed a typo, which we fixed in the new version.
>
> * First we have provided an interval of validity for $\alpha$ and $\beta$. However, we felt that working with a fixed, reasonable number will increase the readability. Following the comments of the Referee we have changed this back and provide a validity interval.
> * Similarly, for the quantities depending on $\alpha$ or $\beta$ (e.g. $\kappa_{\alpha}$) we have considered a fixed value $\alpha$ to improve readability. But as above, they can be given on an interval as well, if requested.
>
> * Following the Referee's suggestion, in the new version, to increase the self containedness and improve the readability, we have added the frequently refereed lemmas and theorem to the appendix.
> * We apologise for the error encountered while running the R script. It is caused by a syntax/undefined object related error by a line that is meant to exemplify. The part of the code that runs the simulation (from the "#SIMULATION" comment downwards) should run without error.
>
> * We have extended the explanation of the simulation study and described its connection and relevance to the theoretical results in more detail. Indeed, the first message is that meta learning is substantially better than using just one experiment. The second is that combing $\chi^2$ statistics is substantially worse than computing the $\chi^2$ test using all the data. Finally, we show that depending on the interplay of $m$ and $d$ either the standard $\chi^2$ combination or the novel directional test statistics method is better.
>
> * We note, that the ROC curve indicates the trade-off between sensitivity and specificity for each of the tests, hence describe the precision of the test.
>
>  * Following the suggestion of the Referee, we have extended the simulation section, see the attached figures in the general rebuttal.

---

> > ### Comment · Reviewer_qxf1 · 2023-08-14
> >
> > I have read the response of the authors and thank them for considering my remarks and those of my fellow reviewers.
> >
> > As for the R script, when I run the code from the "#SIMULATION" comment downwards as suggested by the authors, I still get the same error.
> >
> > Overall, I wish to maintain my score of 6.

---

### Official Review · Reviewer_aj9v · 2023-07-31

**Soundness:** 3 good
**Presentation:** 3 good
**Contribution:** 3 good
**Rating:** 6
**Confidence:** 4

**Summary:**

The paper considers methods to aggregate test statistics from different, independant, sources, in order to construct an aggregated test with hopefully more power. The key contribution of the paper is the study of the minimal treatment effect which can be detected in a standard gaussian noise setting, for which they obtain minimax rates.
These rates exhibit an elbow effect when the number of aggregated statistics (m) is close to the square of the dimension of the signal (d^2), which the authors relate to the use of the signal direction in the test (when $m < d^2$, the standard chi2 test would give near optimal power, while for $m > d^2$, the test statistics must encode directional information if optimal power is to be obtained). These rates bring two insights: First, aggregating one dimensional statistics in a multi dimensional setting comes at a price. Second, there is no single optimal aggregating method.

**Strengths:**

Overall, the paper is well presented and obtains conclusive results in the scope considered, in the form of minimax rates. These rates justify previous empirical insights on aggregated testing strategies, notably the need for different aggregating strategies depending on the number of tests and the dimension of the problem. Methods achieving the rates (up to a log factor) are specified.

As far as I could assess, the mathematical proofs are, up to small typos (see weakness), correct. The presentation of the main results in section 2 can be easily followed (minimax rates in the general case, optimal combination methods then improved minimax rate using coordination between tests).



**Weaknesses:**

The proof in the appendix suffers from some small typos. Notably, I believe that in equation (S.1), the $2\epsilon$ term should be $\epsilon$ (or $\epsilon< \frac{1}{2}\left(\kappa_{1/10} - \kappa_{1/8}\right)$ in the definition of $\epsilon$), while in line 538, the conclusion of Markov's inequality is that $D^c$, not $D$, has mass less than $1/64$.

The methodology used to obtain Figure 1. could be improved. Notably, the Roc Curves for Chi-square combined and Chi-square pooled should not exhibit any randomness, since these two curves can be computed in closed form using the cumulative distribution functions of the chi2 square and non central chi square. If numerical approximations are to be used, it could be possible to obtain curves exhibiting much less noise by increasing the number of repeats and recycling them for all FPR (I could obtain curves exhibiting little to no noise robustly using 10 000 repeats and 100 FDR in less than <2s on my personal computer, so computation time is not an issue).
Moreover, the way the $f_i$ are drawn, using Rademacher random variables, might have an impact on the directional methods.  While this might or might not be the case, I would suggest recomputing the curve, drawing a random f uniformly on the sphere.

**Questions:**

Could the methodology be extended beyond the current setting? Notably, is there any natural generalisation in the case where the noise level $\sigma = 1/\sqrt{n}$ can no longer be assumed to be identical ?

Is there any explanation of the results in terms of the distribution of the p-values under the alternative hypothesis? If so, is there any insight on the best way to aggregate a given set of statistics (instead of considering the best way to aggregate the best statistics for a given m, d)?

**Limitations:**

The paper derives optimal test aggregation strategies in the context of gaussian noised signals. A first limitation of the paper is that it is assumed that the sample size $n$ considered in each collected test is identical. This can barely be expected in practice, and as such, insight about the impact of uneven tests would be welcome (i.e., does $mn$ translate into $\sum_{i=1}^m n_i$?, or rather $m \min(n_i)$?). This issue is not mentionned in the paper.

Another limitation not mentionned is the fact that, in practice, the test statistics obtained from independent trials are not chosen, but set, and as such, Stouffer's method, which attain the minimax rates when $m>d^2$, is not implementable. In most settings where it could be implemented, the whole $X_i$ information would be known, and therefore the dimension reduction issue would not occur. For this reason, the best rate achievable with realistic one dimensional statistics is of particular interest. Unfortunatly, this is left to the appendix, in Theorem 4.

---

> ### Author Rebuttal · Authors · 2023-08-09
>
> We express our sincere thanks to the Reviewer for the taking the time and effort to thoroughly review, the insightful comments and constructive feedback on our paper. The Reviewer also identifies areas for improvement, which we will address point-by-point below.
>
> *Typos:* Thank you for pointing out these typos, we have corrected them in the revised manuscript.
>
> *Simulation study:* we have rerun the simulations with a much larger number of repeats (i.e. 100 000 repeats) and considered a higher dimensional setting. Furthermore, following the suggestion of the Referee, we have drawn the parameter $f$ uniformly from the sphere. See the attached figures in the general rebuttal. We note that we did not observe a significant difference compared to the Rademacher prior, but we if it is deemed beneficial we can include this in the simulations added to the appendix of the new version of the article.
>
> *Optimal test statistic:* We  agree that in case of independent trials typically the test statistics are set independently from the number of trials. Hence in such cases the direction based approach can not be applied and hence the best attainable rate is $\frac{\sqrt{d}}{mn}$ following Theorem 4. We completely agree with the Reviewer, that this result is of particular interest in such settings. In our revised manuscript we have highlighted these more prominently, stressing the importance of theoretically capturing explicitly the loss in terms of error rate (i.e. the $\sqrt{m}$ worse error rate) when conducting meta-analysis in such scenarios. It is important for practitioners conducting meta-analysis to be aware of this potential loss in statistical power. and provided an extended description of Theorem 4. Nevertheless, one could also consider situations where the number of trials is known in advance or a new round of experiments are executed in addition to some available earlier ones. In such cases a better rate is attainable by choosing the non-standard, directional test statistics is designed for scenarios where individual datasets are not centrally collected, but there's coordination among experimenters (depending on the interplay between m and d).
>
> *Concerning the heterogeneity and different sample sizes:* In the revised paper have included a discussion concerning extensions in this direction. We note that in case the sample sizes are of the same order, i.e.  $n_j \asymp n_k$ for all $k,j=1,\dots,m$, the current rates can be proven with minor modifications. If the local sample sizes can be substantially different then the question is more complicated. The rate $m {\min}_j n_j$ can be of course easily obtained, but it is rather pessimistic, e.g. if $n_1\gg\underset{j=2}{\overset{m}{\sum}} n_j$ then almost no extra information comes from the experiments $j=2,...,m$ and the rate is determined by the first sample size $n_1$, which is substantially better than $m {\min}_j n_j$. However, the rigorous analysis of this question goes beyond the scope of our paper.
>
> *Question concerning insight on how to aggregate:* This is a great question, and something that was not highlighted in the old version of the paper. The insight is mainly in the fact that in terms of error rate, many methods perform optimally in principle, as long as the underlying p-value / test statistics have decent power. That is, Fisher's method, Stouffer's or simply averaging all can reach the $\frac{\sqrt{d}}{\sqrt{m}n}$ optimal rate. Beyond that, simulation of the behaviour p-values / test statistics for specific alternatives that are especially deemed important to detect is also an option. We have added a remark reflecting this to the new version of the paper.

---

> > ### Comment · Reviewer_aj9v · 2023-08-18
> > **Answer to rebuttal**
> >
> > I have read the authors' response and thank them for taking the remarks into consideration.
> >
> > For the simulation study:
> > Thank you for rerunning the simulations, the new plots are cleaner and easier to interpret. The authors have thoroughly answered my concern about the potential bias due to the Rademacher prior.
> >
> > Concerning the heterogeneity and different sample sizes:
> > Including a discussion in the revised paper is sufficient, and I agree with the authors that a rigorous analysis of the setting where $m \min_j n_j \ll \sum_j n_j$ is beyond the scope of the present paper.
> >
> >
> > All in all, the authors' answers were satisfactory, as they cleared up a potential weakness and added discussions on potential extensions of their work. The paper is technically solid and well presented. Overall, I wish to maintain my score to 6.

---

### Author Rebuttal · Authors · 2023-08-09

First of all we would like to thank the Reviewers for carefully reading our paper and their interest in our work. We are happy to hear that the majority of the reviewers found our paper "well presented" (aj9v), "well-written and organised" (qxf1),  written in a clear and without excessive statistical jargon (vLRc), "extremely well written" and has "textbook" quality" (ThPF). We also thank that the reviewers found our work "original" (ThPF, vLRc) with "sound theoretical results" (GjrH)
"up to small typos, correct mathematical proofs" (aj9v) and that "The theory and experiments are in perfect accordance and complement each other."

The referees have also raised a few concerns and provided several suggestions which we have addressed point-by-point in the individual rebuttals. Here we collect the main changes in the manuscript.

* We have extended the discussion of our results and numerical analysis following the suggestions of the referees (see the comments below for details). We have also corrected the typos pointed out by the referees.
* We have clarified the use of misinterpretable jargons to further improve the readability.
* We have extended the simulation study, in the new experiment we use substantially more repetitions resulting in a more accurate picture with negligible randomness visible.
* We have expanded on the relevance of our work to the field of machine learning.
* We have added references to existing literature on multiple testing where error rate is of concern.
* We provide a more extensive description the model and its connection to other, practically more important statistical models, whilst also emphasising the limitations of this model.
* We added more intuitive explanation as well as interpretation of which method / regime applies in practice, depending on the experiment design.

We would like to thank all the Reviewers for their consideration.

---

### Decision · Program_Chairs · 2023-09-21

**Decision:**

Accept (poster)

**Comment:**

This meta review is based on the reviews, the authors rebuttal and the discussions with the reviewers, discussions with the SAC, and ultimately my own judgement on the paper. There was a consensus that the paper contributes sound and interesting contributions to the theoretical analysis of decision making. I feel this work deserves to be featured at NeurIPS and will attract interest from the community. I would like to personally invite the authors to carefully revise their manuscript to take into account the remarks and suggestions made by reviewers. Congratulations!